# Modeling of Equivalent Circuit Analysis of Degraded Electric Double-Layer Capacitors

**DOI:** 10.3390/ma14020435

**Published:** 2021-01-17

**Authors:** Tomoki Omori, Masahiro Nakanishi, Daisuke Tashima

**Affiliations:** 1Department of Electrical Engineering, Graduate School of Engineering, Fukuoka Institute of Technology, Wajiro-higashi 3-30-1, Higashi-ku, Fukuoka 811-0295, Japan; mem18105@bene.fit.ac.jp; 2Department of Electrical Engineering, Faculty of Engineering, Fukuoka Institute of Technology, Wajiro-higashi 3-30-1, Higashi-ku, Fukuoka 811-0295, Japan; m-nakanishi@fit.ac.jp

**Keywords:** supercapacitor, electric double-layer capacitor, alternating current impedance method, equivalent circuit analysis, modeling

## Abstract

The demand for electric double-layer capacitors (EDLCs) has recently increased, especially for regenerative braking systems in electric or hybrid vehicles. However, using EDLCs under high temperature often enhances their degradation. Continuously monitoring EDLC degradation is important to prevent sudden malfunction and rapid drops in efficiency. Therefore, it is useful to diagnose the degradation at a lower frequency than that used in charge/discharge. Unused and degraded EDLCs were analyzed using the alternating current impedance method for measurements over a wide frequency range. Each result had a different spectrum up to 1 kHz. In addition, we show the basic inside condition of EDLCs with equivalent circuit analysis. This paper explores the possibility of degradation diagnosis at a high frequency and the basic physical mechanism.

## 1. Introduction

Electric double-layer capacitors (EDLCs) are energy storage devices that store energy by forming an electric double layer between a porous electrode and an electrolyte [1,2]. EDLCs are characterized by their ability to rapidly charge and discharge. Moreover, they have a long life and are virtually maintenance-free owing to no chemical changes [3,4,5,6]. Due to these advantages, EDLCs are increasingly used as energy storage devices for electric vehicles, wind turbines, and solar power generators [4,5,6,7,8]. There has been an especially dramatic increase in the application of EDLCs to the regenerative energy recovery of electric vehicles.

However, the most serious problem when applying EDLCs to vehicles is their degradation. The degradation of EDLCs is enhanced by harsh environments, such as high temperature and voltage [9,10,11,12,13,14,15], and EDLCs are actually exposed to such environments in vehicles. Therefore, it is necessary to continuously monitor and diagnose the conditions of EDLCs; otherwise, sudden malfunctions or an efficiency drop cannot be avoided. 

Measuring the impedance of EDLCs is likely the best technique to monitor their condition as impedance reflects the internal states of devices and can be measured in a straightforward manner. We especially focus on a higher-frequency impedance rather than the timescale of charge and discharge. Electrical phenomena with different timescales can be separated using filter techniques. Thus, a diagnosis circuit at a higher frequency can work independently of charge/discharge circuits operated in a lower frequency range [16]. In addition, a diagnosis circuit at a higher frequency has the advantage of a high diagnostic speed. Therefore, diagnosing the degradation at a high frequency is a groundbreaking method to constantly observe the degradation; however, previous studies have yet to report on these aspects.

To understand the best frequency for diagnostic impedance measurements, we carried out alternating current impedance measurements on unused and degraded EDLCs over a wide frequency range, followed by a comparison of their impedance spectra [17,18]. In addition, equivalent circuit analysis was also conducted to inspect the physical mechanism behind the spectral change between unused and degraded EDLCs, which reinforces the reliability of the diagnosis.

## 2. Materials and Methods 

The EDLCs were measured using the alternating current impedance method. Figure 1a shows the unused EDLCs, which were provided by the Nippon Chemi-Con Company (DLCAP DXE, 2.5 V, 400 F). Two EDLCs were used in this study. One of the EDLCs was measured as received, while the other EDLC was degraded at 60 °C and 3.2 V for 9 weeks. Figure 1b shows the structure of the EDLCs used in this study. The EDLCs were cylindrical and composed of an electrolyte (1 M (C_2_H_5_)_4_NBF_4_/PC), activated carbon electrode, and a separator (cellulose) [19]. Figure 2 shows the circuit for measuring the EDLCs with the alternating current impedance method. The voltages applied to the EDLCs and the currents flowing through the devices were measured by two channels of an oscilloscope (Tektronix TDS 3021D, Tokyo, Japan). The former was amplified by a factor of 100 with a preamplifier (TURTLE T-01LGA, Ibaragi, Japan), whereas the currents were transformed into voltages using a transimpedance amplifier (nF CA5350, Kanagawa, Japan). The voltages applied to the EDLCs were measured on Channel 1, and the currents that flowed into the EDLCs were measured on Channel 2. To prevent the flow of a large current through the transimpedance amp, a shunt was inserted before the transimpedance amp.

The impedance ($\dot{Z}$) was derived from the voltages ($\dot{V}$) and currents ($\dot{I}$) measured on the oscilloscope via Equation (1). The capacitance ($\dot{C}$) was derived from $\dot{Z}$ with Equation (2) as follows:(1)$$\dot{Z}=\frac{\dot{V}}{\dot{I}}\;\mathrm{and}$$
(2)$$\dot{C}=\frac{1}{j\omega \dot{Z}}$$

## 3. Results

### 3.1. Equivalent Circuit Analysis of an EDLC

Figure 3 shows the measurement results for the unused and degraded EDLCs using the alternating current impedance method. The unused EDLC capacitance measurements showed a significant increase in the low-frequency region (10^–3^ to 10 Hz) and a maximum value of approximately 10^–2^ Hz. In contrast, the degraded EDLC results were different from those of the unused EDLC results in that there was a gradual decrease detected in the range of 10^–3^ to 100 Hz. In addition, no maximum value was observed at approximately 10^–3^ Hz. As the capacitance increased, a peak likely existed below 10^–3^ Hz, which was close to the value of the unused EDLCs results. While the degraded capacitance was totally unchanged, a conclusive difference in the peaks was identified. This difference caused a significant change in the charge/discharge rate, which is an advantage of EDLCs. The rated capacity of this type of EDLC is 400 F, but the measured maximum capacity of the unused EDLC was approximately 270 F. This discrepancy can be attributed to the voltage dependence of the EDLCs. In particular, EDLCs have a fractal structure [20] and strong voltage dependence [16,21]. In this experiment, the applied voltage amplitude was 2 V and the impedance of the EDLC was very low; therefore, the voltage was not applied adequately. This was the reason why the measurement results did not reach the rated capacity.

The impedance measurements of the unused EDLCs showed a constant value in the 10^–3^ to 10^3^ Hz frequency range and a decrease at frequencies higher than 10^3^ Hz. In contrast, the degraded EDLC was observed to have a difference in impedance in the low-frequency area, and the impedance was 600 mΩ. There was a gradual decrease with the increasing frequency from 10^–3^ to 10^3^ Hz, and a radical decrease occurred at more than 10^3^ Hz. The rated impedance of the EDLC is 2.5 mΩ, but the measured impedance in the low-frequency range was approximately 40 mΩ. As the impedance of the EDLC was very low, the contact resistance between the terminals affected the measured impedance value. The decrease in resistance that occurred at values less than 10^3^ Hz was the cause of the degradation observed in this study. To analyze this degradation factor in detail, an equivalent circuit analysis was performed by measuring the real and imaginary parts of the impedance, and the internal condition was represented electrically.

Figure 4 and Figure 5 show the results for the equivalent circuit analysis of the unused and degraded EDLCs. The results for the equivalent circuit analysis of the unused EDLCs can be divided into three processes corresponding to the low-, mid-, and high-frequency regions [16]. In the high-frequency region, data indicate that the orientational polarization had an effect. This phenomenon occurred because orientational polarization is a relatively fast polarization reaction, and the results for the capacitance measurements were quite low, i.e., less than 1 mF. In the mid-frequency region, the space charge polarization, which was slower than the orientational polarization, was considered to have exerted an effect. The formation of the electric double layer affected the devices in the low-frequency region. The capacitance measurement results in the low-frequency region show the highest capacitance, which was the closest value to the rated capacitance. The time required for the formation of the electric double layer was longer than that of the other two polarizations. By taking these factors into account, an electrical equivalent circuit model was created. The internal models of the EDLCs and the resistance of the pores are represented electrically with capacitors, resistances, and Warburg impedances [9,22,23,24,25]. Figure 6a shows the created equivalent circuit model. The impedance, $Z_{3}$, in the high-frequency region, where the orientational polarization had an effect, is represented by the parallel circuit of a capacitor ($C_{3}$) and resistor ($R_{3}$). Here, $C_{3}$ represents the action of the oriented polarization, and $R_{3}$ represents the equivalent series and contact resistance. The impedance, $Z_{2}$, in the mid-frequency region is represented by the parallel circuit of a capacitor ($C_{2}$), resistor ($R_{2}$), and constant-phase element ($Z_{CPE2}$). Here, $C_{2}$ represents the action of space charge polarization, and $R_{2}$ represents the solution resistance. Furthermore, $Z_{CPE2}$ represents the fractal structure of the EDLC, as well as the relationship between the pores on the electrode surface and electrolyte. The impedance, $Z_{1}$, in the low-frequency region is represented by a capacitor, $C_{1}$. Here, $C_{1}$ affects the formation of the electrical double layer. This unused EDLC impedance, $Z_{U}$, is expressed by the following equation:(3)$$Z_{U}=Z_{1}+Z_{2}+Z_{3}$$

Here, $Z_{1}$, $Z_{2}$, and $Z_{3}$ can be represented as follows:(4)$$Z_{1}=\frac{1}{j\omega C_{1}},$$
(5)$$Z_{2}=\frac{R_{2}Z_{CPE2}}{R_{2+}Z_{CPE2}\left(1+j\omega C_{2}R_{2}\right)},\;\mathrm{and}\;$$
(6)$$Z_{3}=\;\frac{R_{3}}{1+j\omega C_{3}R_{3}}$$

In addition,$\;Z_{CPE}$ can be expressed as follows:(7)$$Z_{CPE2}=R_{0}{\left(j\omega \tau_{0}\right)}^{-\alpha },$$
where $R_{0}$ and $\tau_{0}$ are constant and α is set between 0 and 1. Equation (3) can be expressed as follows:(8)$$Z_{U}=\frac{1}{j\omega C}+\frac{R_{2}R_{0}{\left(j\omega \tau_{0}\right)}^{-\alpha }}{R_{2+}R_{0}{\left(j\omega \tau_{0}\right)}^{-\alpha }\left(1+j\omega C_{2}R_{2}\right)}+\frac{R_{3}}{1+j\omega C_{3}R_{3}}$$

Figure 6b shows the equivalent circuit model of the degraded EDLC. As mentioned previously, there was a notable difference between the impedance of the unused and degraded EDLCs in the low- to mid-frequency region, and this factor played a role in the degradation. The degradation factor can be expressed by a new constant-phase element, i.e., $Z_{CPE4}$. The $Z_{CPE4}$ parameter represents the slope of the impedance of the degraded EDLC in the low- to mid-frequency region, and the position of $Z_{CPE4}$ is between $Z_{1}$ and $Z_{2}$. The degraded EDLC impedance, $Z_{D}$, can be expressed as follows:(9)$$Z_{D}=\frac{1}{j\omega C}+Z_{CPE4}+\frac{R_{2}R_{0}{\left(j\omega \tau_{0}\right)}^{-\alpha }}{R_{2+}R_{0}{\left(j\omega \tau_{0}\right)}^{-\alpha }\left(1+j\omega C_{2}R_{2}\right)}+\frac{R_{3}}{1+j\omega C_{3}R_{3}}$$

Here, $Z_{CPE4}$ can also be represented as follows:(10)$$Z_{CPE4}=R\prime_{0}{\left(j\omega \tau_{0}{}^{\prime }\right)}^{-\alpha^{\prime }}\;$$

Therefore, $Z_{D}$ is as follows:(11)$$Z_{D}=\frac{1}{j\omega C}+R\prime_{0}{\left(j\omega \tau_{0}{}^{\prime }\right)}^{-\alpha^{\prime }}+\frac{R_{2}R_{0}{\left(j\omega \tau_{0}\right)}^{-\alpha }}{R_{2+}R_{0}{\left(j\omega \tau_{0}\right)}^{-\alpha }\left(1+j\omega C_{2}R_{2}\right)}+\frac{R_{3}}{1+j\omega C_{3}R_{3}}$$

### 3.2. Internal Model of EDLC Degradation

Figure 7 shows the internal model of the cathode side of the degraded EDLC, illustrating the equivalent circuit model proposed in the previous section. The models of the unused and degraded EDLCs were created in the previous section, and the location of the degradation was identified. As the degradation element, i.e., $Z_{CPE4}$, was located between $Z_{1}$ and $Z_{2}$, degradation was considered to occur between the electrode surface and electrolyte inside the degraded EDLC. Moreover, as the degradation of the EDLC was greater on the cathode side than on the anode side, the degradation of the EDLC was considered to be dependent on the cathode side [26,27,28,29,30,31]. The degradation tests showed that pores were the main degradation sites. The degradation factors were identified as gas generation [32,33,34] and pore reduction [2,26,27,30,35,36] due to chemical reactions. In this experiment, gas generation due to the high temperature increased the internal pressure and amount of water. In addition, the high voltage may have caused the pores to melt and shrink. In either case, the internal condition of the EDLC should be observed to identify the causes of degradation.

## 4. Conclusions

In this study, the impedances of degraded and unused EDLCs were measured over a wide frequency range. Comparing each impedance measurement, there were differences of up to 1 kHz. Therefore, our method is able to diagnose degradation in the high-frequency range, which is lower than the frequency range used for charge/discharge. 

In addition, equivalent circuits were created from these measurements, allowing us to depict the basic data within the EDLCs. Comparing the basic data to the degraded EDLCs, we were able to examine the degradation factor. 

Our results allow the possibility of diagnosing degradation in the high-frequency range and internal conditions of the EDLCs. Future studies should focus on creating a degradation diagnostic circuit and observing the surface of the active carbon electrode.

## Figures and Tables

**Figure 1 materials-14-00435-f001:**
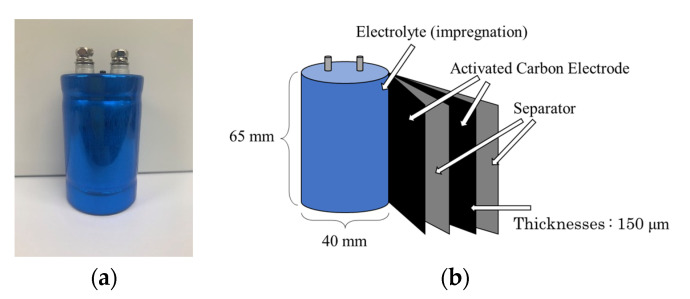
(**a**) Investigated electric double-layer capacitors (EDLCs) and (**b**) their configuration.

**Figure 2 materials-14-00435-f002:**
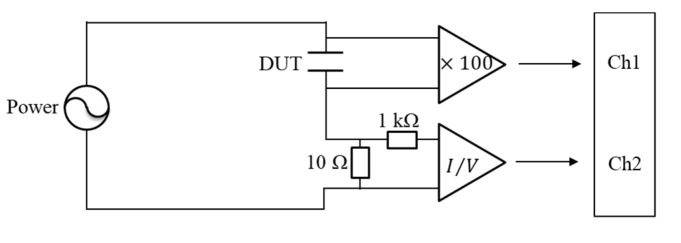
Measuring circuit for the EDLCs.

**Figure 3 materials-14-00435-f003:**
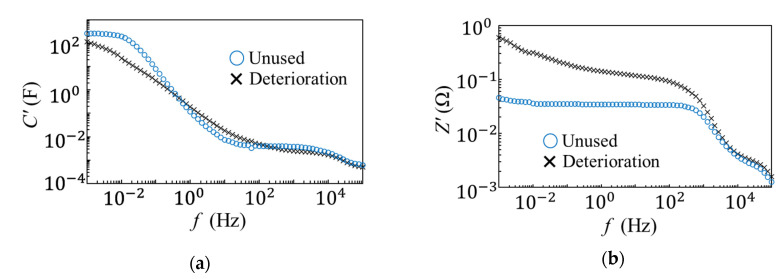
Capacitance and impedance measurements of the unused and degraded EDLCs. (**a**) Capacitance measurements and (**b**) impedance measurements.

**Figure 4 materials-14-00435-f004:**
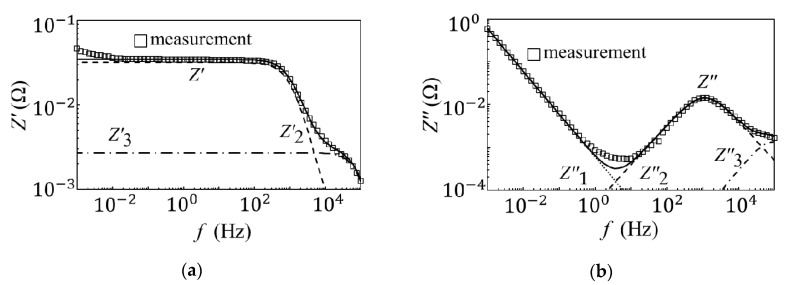
The equivalent circuit analysis of the unused EDLCs. (**a**) The equivalent circuit analysis results for the real part impedance of the unused EDLCs. (**b**) The equivalent circuit analysis results for the imaginary part impedance of the unused EDLCs.

**Figure 5 materials-14-00435-f005:**
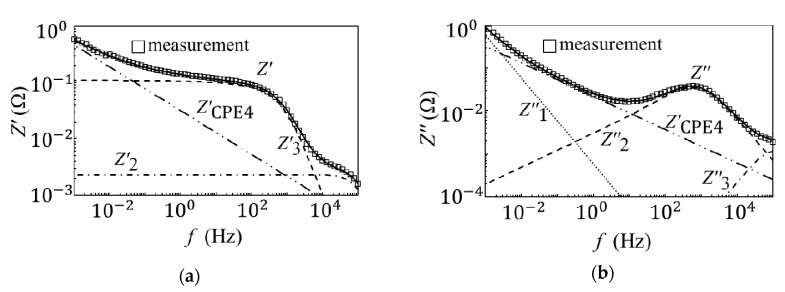
The equivalent circuit analysis of the degraded EDLCs. (**a**) The equivalent circuit analysis results for the real part capacitance of the degraded EDLCs. (**b**) The equivalent circuit analysis results for the imaginary part capacitance of the degraded EDLCs.

**Figure 6 materials-14-00435-f006:**
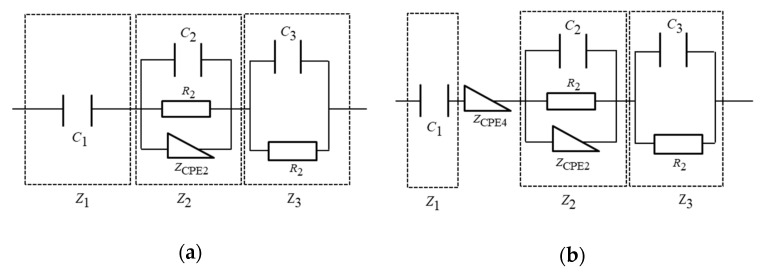
Equivalent circuit model. (**a**) Equivalent circuit model of the unused EDLC. (**b**) Equivalent circuit model of the degraded EDLC.

**Figure 7 materials-14-00435-f007:**
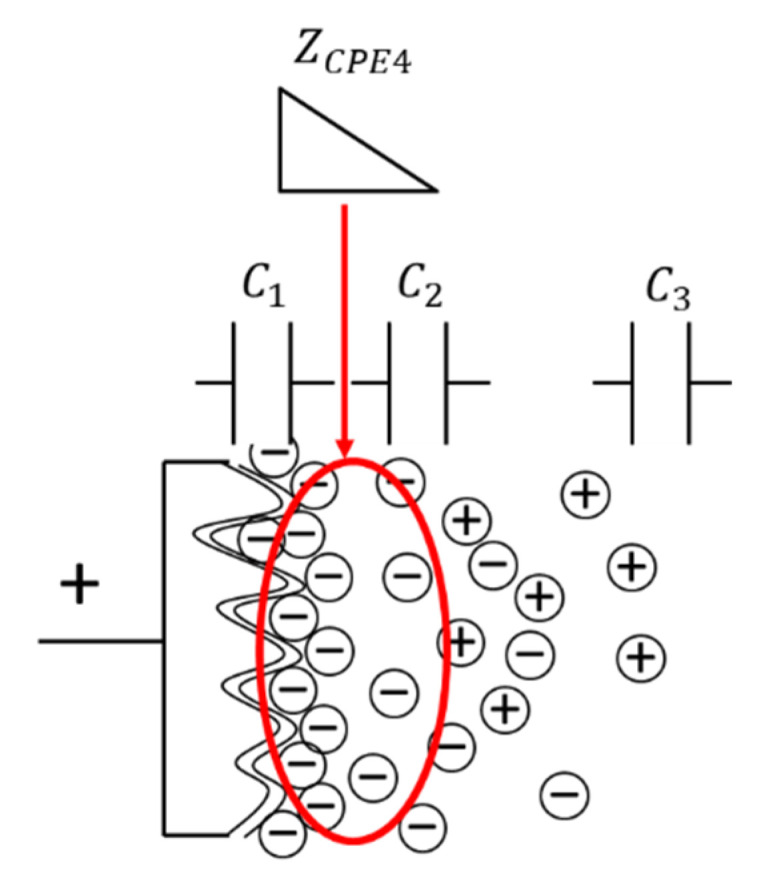
Internal model of the cathode side of a degraded EDLC.

## Data Availability

Data sharing not applicable to this article as no datasets were generated.

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
