# Peer review of "Modeling of Equivalent Circuit Analysis of Degraded Electric Double-Layer Capacitors"

_materials, 2021, doi:10.3390/ma14020435_

Round 1

Reviewer 1 Report

The paper presents studies monitoring of degradation of electric double layer capacitors (EDLCs) through equivalent circuit analysis. There are many interesting results and interpretations made.

However I have found some corrections to be made.

Page 1 row 31  “they are a long-life and virtually” delete a

Page 1 rows 32-34 “They have been used in vehicles to recover regenerative energy and as energy storage devices for renewable energy with these features in recent years [4-8].” Please rephrase.

Page 1 row 34 “On the other hand, EDLCs have a low energy density [9, 10].” On the other hand is related to what? Please change.

Page 1 rows 34-35 The proposed solution is to increase the operating voltage,” the proposed solution for what?

Page 1 rows 36-38 “One of the greatest strengths of EDLCs is that they are practically maintenance-free devices, but this is only an ideal feature. In fact, the use of EDLCs in harsh environments is increasing rapidly with increasing in their demand.” The phrases need to be rewritten for better understanding.

Page 1 rows 38-39 “Especially, EDLCs are expected to increase the durability with developing the automobile. the main causes    “ Please rewrite and The main cause  is with capital letter

Page 1 rows 42-43 Therefore, the degradation behavior of EDLCs was modeled electrically and the identification of degradation areas and causes of degradation are discussed in this study. In addition Page 2 rows 44-46 “the purpose of this study was to diagnose degradation at high frequency. The diagnosis of degradation at high frequency is considered very efficient because it can be performed independently of the low frequency region where EDLC is usually used [19].” In my opinion the text does not explain very clear the purpose.

My opinion is that the Introduction must be rewritten.

In my opinion Figures 1 and 2 can be unified.

Please explain why from 4 EDLCs only 1 was degraded. Why have you choose and didn’t degrade 2 or 3 of them and compare the results. You did not discuss the measurements made on the 3 unused EDLCs.

Page 2 row 58 “The voltages applied to the ELDCs” replace with EDLCs

Page 2 rows 60-61 “the latter were transformed into voltages by using a transimpedance amplifier” you can replace by “the currents were transformed into voltages by using a transimpedance amplifier”

Page 2 row 75 you did not explained the terms from eq 2.

Page 2 rows 78-79 “The results for unused and degraded EDLC measurements derived by using the AC impedance method are shown in Figure 4.” measurements derived? please rephrase

Page 2 row 80 (??? to 10 Hz) and a peak around ??? Hz from Fig. 4 it does not look like a peak, it is rather a plateau

Page 2 rows 86-88 “The rated capacity of this type of EDLC is 400 F, but the measured maximum capacity of the unused EDLC was approximately 270 F.” If the producer Nippon Chemi-Con Company (DLCAP DXE, 2.5 V, 400 F), declared the capacity of 400 F and the measurements showed only 270 F maybe the measurements were not made properly.

Page 2 rows 95-96 “Impedance measurements of the unused EDLCs were constant from ??? to ??? Hz and decreased with the increasing frequency at values of more than ??? Hz.” In my opinion you can better express as The impedance measurements of the unused EDLCs showed a constant value in the ??? to ??? Hz frequency range and a decrease at frequencies higher than ??? Hz.

Page 4 row 136 What is R0?

Page 6 rows 167-168 “Some degradation tests have been conducted, and the main degradation sites have been reported to be pores.” My suggestion is to change with The degradation tests made showed that the main degradation sites were the pores.

Page 6 rows 169-170 “In addition, the degradation factors were identified as gas generation [34-36] and pore reduction” My suggestion is to delete In addition

The Conclusions should be rewritten.

Reviewer 2 Report

The paper has potential to be very nice, there are a few things that need to be tidied up however.

Abstract - the key to a good abstract is that it should tell prospective readers what to expect from the paper, this is missing. Add a final line e.g. "This paper explores a model for the degradation in capacitance for EDLCs"

Lines 14 (and 29 etc) EDLC generally stands for Electrostatic rather than Electric, the former being more descriptive separating from electrochemical types.

line 35 - change is to are

line 37/38 - poor english, reword or remove (does it add anything?)

line 39 - capitalise t in The

line 48 - Alternating not alternative

line 48 - (AC) - remove this is really too common to spell out

line 56 - electrolyte - which?

line 56 - separate - what material, these both are critical to degradation temperature and voltage.

line 56 - activated carbon electrode, this is common however manufacture datasheet says aluminium, which is it?

Figure 1 heading - "Unused" - why does this matter for the photo, suggest remove this word

Figure 2 - manufacturers datasheet has more information than this, also this figure tells the reader nothing he doesn't know. Please add a lot more information (materials thicknesses etc) otherwise what does it help with?

Line 80,82,83 - unbold the 10 - there is no ambiguity in the context (suggestion)

line 89 - fractal - agreed but a reference here would be useful

line 178 - temperature - I missed these results in the earlier sections, perhaps be more clear where or add more temperature results to emphasise

References - loads here, but I don't see the manufacturers own datasheet  -why not? 

It clearly states the degradation to 70% after 2000hrs at 70ºC with DC, an excellent reference point I'd suggest.

Reviewer 3 Report

In this paper, Tomoki et.al. proposed an equivalent circuit model to explain the degradation of electric double layer capacitors (EDLC). The authors made use of the frequency dependent impedance in their model to model the behaviour of unused EDLC and degraded EDLC.

The abstract is poorly written and lack in objective/direction. The initial part of the introduction is reasonable, but the latter part of the introduction again shows lack of clear direction (i.e. the purpose of the study is to diagnose degradation at high frequency, but please elaborate more on why is it important?).

"In contrast, the 80 degraded EDLC results were different from the unused EDLC results in that there was a gradual increase detected in the range of ??−? to 100 Hz." there seems to be a mistake, it should be gradual decrease? 

In Figure 4b, the authors did not explain why the impedance is higher in degraded EDLC.
The resolution for figure 5a is too blur.

Overall, there is no clear presentation on how the model can be used to monitor degradation in EDLC. The manuscript is clearly lacking in the standards to be published in materials. I recommend rejection.

Reviewer 4 Report

The authors report on circuit analysis of electric double layer capacitors (EDLC) degradation and the results of EDLC were partially frequency-dependent. While the paper is well written, it is difficult for me to see what contribution it makes to the research community. Simply analysis of EDLC degradation and modeling does not seem like much of a contribution unless something about the new design or modeling sheds new light on the problem of EDLC. However, this paper does not seem to make such a contribution. The reviewer thinks that the paper is still on the conference or proposal level.  The authors need to provides a more thorough analysis of the proposed works. Due to the lack of novelty in this work, I do not recommend the manuscript for publication.

Round 2

Reviewer 3 Report

The authors have made a good effort to improve the manuscript. It can now be accepted in the current form.

Most of the concerns are addressed satisfactorily.

Author Response

Comment: English language and style are fine/minor spell check required.

Reply: According to your comment, we revised Zcpe2 and Zcpe2 to ZCPE2.(page 4 lines 139 and 142)

Reviewer 4 Report 

In the revised manuscript, the authors could not address all of my comments with proper explanation. The comments must be well addressed before considering the publication of this manuscript. Here are a few comments that I suggest the author's to address.

  1. The novelty of this work is unclear. In particular, the authors should clarify the novelty of this work.
  2. The reviewer thinks that the paper is still on the conference or proposal level. The authors need to provide a more thorough analysis of the proposed works.

Author Response

Comment:In the revised manuscript, the authors could not address all of my comments with proper explanation. The comments must be well addressed before considering the publication of this manuscript. Here are a few comments that I suggest the author's to address. The novelty of this work is unclear. In particular, the authors should clarify the novelty of this work. The reviewer thinks that the paper is still on the conference or proposal level. The authors need to provide a more thorough analysis of the proposed works.

Reply: According to your comments, we have modified the introduction and conclusions such that the novelty and relevance/contribution of this study are clearer. (page 2 lines 49; page 7 lines 192)